# Ranking Policy Decisions

**Hadrien Pouget**[*]
University of Cambridge
UK
pougeth@gmail.com

**Hana Chockler**
causaLens
and
King's College London
UK
hana@causalens.com
hana.chockler@kcl.ac.uk

**Youcheng Sun**
Queen's University Belfast
UK
youcheng.sun@qub.ac.uk

**Daniel Kroening**[†]
Amazon
UK
daniel.kroening@magd.ox.ac.uk

## Abstract

Policies trained via Reinforcement Learning (RL) are often needlessly complex, making them difficult to analyse and interpret. In a run with $n$ time steps, a policy will make $n$ decisions on actions to take; we conjecture that only a small subset of these decisions delivers value over selecting a simple default action. Given a trained policy, we propose a novel black-box method based on statistical fault localisation that ranks the states of the environment according to the importance of decisions made in those states. We argue that among other things, the ranked list of states can help explain and understand the policy. As the ranking method is statistical, a direct evaluation of its quality is hard. As a proxy for quality, we use the ranking to create new, simpler policies from the original ones by pruning decisions identified as unimportant (that is, replacing them by default actions) and measuring the impact on performance. Our experiments on a diverse set of standard benchmarks demonstrate that pruned policies can perform on a level comparable to the original policies. Conversely, we show that naive approaches for ranking policy decisions, e.g., ranking based on the frequency of visiting a state, do not result in high-performing pruned policies.

## 1   Introduction

Reinforcement learning is a powerful method for training policies that complete tasks in complex environments. The policies produced are optimised to maximise the expected reward provided by the environment. While performance is clearly an important goal, the reward typically does not capture the entire range of our preferences. By focusing solely on performance, we risk overlooking the demand for models that are easier to analyse, predict and interpret [16]. Our hypothesis is that many trained policies are *needlessly complex*, i.e., that there exist alternative policies that perform just as well or nearly as well but that are significantly simpler. This tension between performance and simplicity is central to the field of explainable AI (XAI), and machine learning as a whole [11]; our method aims to help by highlighting the most important parts of a policy.

---

[*]The work in this paper was done while at the University of Oxford.
[†]The work in this paper was done prior to joining Amazon.

35th Conference on Neural Information Processing Systems (NeurIPS 2021).

The starting point for our definition of "simplicity" is the assumption that there exists a way to make a "simple choice", that is, there is a simple default action for the environment. We argue that this is the case for many environments in which RL is applied: for example, "repeat previous action" is a straightforward default action for navigation tasks.

The key contribution of this paper is a novel method for *ranking policy decisions* according to their importance relative to some goal. We argue that the ranked list of decisions is already helpful in explaining how the policy operates. We evaluate our ranking method by using the ranking to simplify policies without compromising performance, hence addressing one of the main hurdles for wide adoption of deep RL: the high complexity of trained policies.

We produce a ranking by scoring the states a policy visits. The rank reflects the impact that replacing the policy's chosen action by the default action has on a user-selected binary outcome, such as "obtain more than X reward". It is intractable to compute this ranking precisely, owing to the high complexity and the stochasticity of the environment and the policy, complex causal interactions between actions and their outcomes, and the sheer size of the problem. Our work uses *spectrum-based fault localisation* (SBFL) techniques [20, 31], borrowed from the software testing domain, to compute an approximation of the ranking of policy decisions. SBFL is an established technique in program testing for ranking the parts of a program source code text that are most likely to contain the root cause of a bug. This ranking is computed by recording the executions of a user-provided test suite. SBFL distinguishes passing and failing executions; failing executions are those that exhibit the bug. Intuitively, a program location is more likely to be the root cause of the bug if it is visited in failing executions but less (or not at all) in passing ones. SBFL is a lightweight technique and its rankings are highly correlated with the location of the root cause of the bug [31]. We argue that SBFL is also a good fit for analysing complex RL policies.

Our method applies to RL policies in a black-box manner, and requires no assumptions about the policy's training or representation. We evaluate the quality of the ranking of the decisions by the proxy of creating new, simpler policies (we call them "pruned policies") without retraining, and then calculate the reward achieved by these policies. Experiments with agents for MiniGrid (a more complex version of gridworlds) [7], CartPole [4] and a number of Atari games [4] demonstrate that pruned policies maintain high performance (similar or only slightly worse than that of the original policy) when taking the default action in the majority of the states (often $90\%$ of the states). As pruned policies are much easier to understand than the original policies, we consider this an important step towards explainable RL. Pruning a given policy does not require re-training, and hence, our procedure is relatively lightweight. Furthermore, the ranking of states by itself provides important insight into the importance of particular decisions for the performance of the policy overall.

The code for reproducing our experiments is available on GitHub[3], and further examples are provided on the project website[4].

## 2   Background

### 2.1   Reinforcement learning (RL)

We use a standard reinforcement learning (RL) setup and assume that the reader is familiar with the basic concepts. An *environment* in RL is defined as a Markov decision process (MDP) and is denoted by $\langle S, A, P, R, \gamma, T \rangle$, where $S$ is the set of states, $A$ is the set of actions, $P$ is the transition function, $R$ is the reward function, $\gamma$ is the discount factor, and $T$ is the set of terminal states. An agent seeks to learn a policy $\pi : S \rightarrow A$ that maximizes the total discounted reward. Starting from the initial state $s_0$ and given the policy $\pi$, the state-value function is the expected future discounted reward as follows:

$$V_\pi(s_0) = \mathbb{E}\left(\sum_{t=0}^{\infty} \gamma^t R(s_t, \pi(s_t), s_{t+1})\right) \tag{1}$$

A policy $\pi : S \rightarrow A$ maps states to the actions taken in these states and may be stochastic. We treat the policy as a black box, and hence make no further assumptions about $\pi$.

---

[3]https://github.com/hadrien-pouget/Ranking-Policy-Decisions. Experiments done at commit c972414
[4]https://www.cprover.org/deepcover/neurips2021/

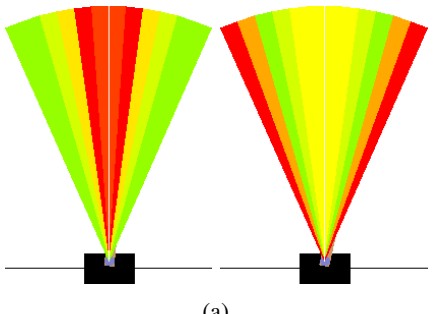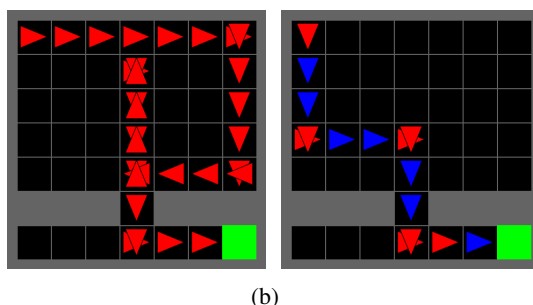

(a)                                                (b)

Figure 1: **(a)** CartPole, in a state where the cart and pole are moving rapidly. The heatmap represents the frequency of appearance of the possible pole angles (left) and the importance scores following SBFL (right). While it is more frequent for the pole to be centred, SBFL successfully identifies that the policy's decisions are more important when the pole is close to falling. **(b)** MiniGrid. Traces of executions with the original policy (left) and a *pruned policy* (right). States in which we take the default action are in blue. Both policies succeed, but pruning unimportant actions simplifies the policy.

## 2.2 Spectrum-based fault localization (SBFL)

The reader is likely less familiar with spectrum-based fault localization (SBFL), as to the best of our knowledge, it has not yet been used in RL. We therefore give a detailed description. SBFL techniques [31] have been widely used as an efficient approach to aid in locating the causes of failures in sequential programs. SBFL techniques rank program elements (say program statements) based on their *suspiciousness scores*, which are computed using correlation-based measures. Intuitively, a program element is more suspicious if it appears in failed executions more frequently than in correct executions, and the exact formulas differ between the measures. Diagnosis of the faulty program can then be conducted by manually examining the ranked list of elements in descending order of their suspiciousness until the cause of the fault is found. It has been shown that SBFL techniques perform well in complex programs [2].

SBFL techniques first execute the program under test using a *test suite*. A test suite comprises of a set of inputs and an expected output for each input. A test *passes* when the output produced by the program under test matches the expected output given by the test suite, and has *failed* otherwise. In addition to the outcome of the test, SBFL techniques record the values of a set of Boolean flags that indicate whether a particular program element was executed by that test.

The task of a fault localization tool is to compute a ranking of the program elements based on the values of the Boolean flags recorded while executing the test suite. Following the notation from [20], the suspiciousness score of each program statement $s$ is calculated from a set of parameters $\langle a_{ep}^s, a_{ef}^s, a_{np}^s, a_{nf}^s \rangle$ that give the number of times the statement $s$ is executed ($e$) or not executed ($n$) on passing ($p$) and on failing ($f$) tests. For instance, $a_{ep}^s$ is the number of tests that passed and executed $s$.

Many measures have been proposed to calculate the suspicious scores of program elements. In Equations (2a)∼(2d) we list a selection of popular and high-performing measures [1, 9, 15, 30]; these are also the measures that we use in our ranking procedure.

$$\text{Ochiai: } \frac{a_{ef}^s}{\sqrt{(a_{ef}^s + a_{nf}^s)(a_{ef}^s + a_{ep}^s)}} \quad \text{(2a)} \qquad \text{Tarantula: } \frac{\frac{a_{ef}^s}{a_{ef}^s + a_{nf}^s}}{\frac{a_{ef}^s}{a_{ef}^s + a_{nf}^s} + \frac{a_{ep}^s}{a_{ep}^s + a_{np}^s}} \quad \text{(2b)}$$

$$\text{Zoltar: } \frac{a_{ef}^s}{a_{ef}^s + a_{nf}^s + a_{ep}^s + \frac{10000 a_{nf}^s a_{ep}^s}{a_{ef}^s}} \quad \text{(2c)} \qquad \text{Wong-II: } a_{ef}^s - a_{ep}^s \quad \text{(2d)}$$

SBFL-based tools present the list of program elements in descending order of their suspiciousness scores to the user. There is no single best measure for fault localization; different measures perform better on different types of programs, and it is best practice to use multiple measures [18]. In our experiments, we combine the four measures listed above for this very reason.

While more sophisticated versions of SBFL exist [3, 5], in this work we prefer to stick to the simpler approach, which was sufficient for producing notable results.

# 3 Method: ranking policy decisions using SBFL

Inspired by the use of SBFL for localising the cause of a program's outcome, we propose a new SBFL-based method to identify the states in which decisions made by an RL policy are most important for achieving its objective. Our method is modular and is composed of two phases: (1) generating mutant policies and (2) ranking states based on the importance.

## 3.1 Definitions

**Executions of RL policies** We apply the SBFL technique to a set of *executions* (sometimes called *trajectories* in the literature) of a given RL policy $\pi$ with mutations. An *execution* $\tau$ of $\pi$ describes a traversal of the agent through the environment MDP using the RL policy $\pi$ and is defined as a sequence of states $s_0, s_1, \ldots$ and actions $a_0, a_1, \ldots$, where $s_0$ is an initial state and each subsequent state $s_{i+1}$ obtained from the previous state $s_i$ by performing action $a_i$, as chosen using $\pi(s_i)$. The last state must be a terminal state. As $\pi$ or the environment can be stochastic, each execution of $\pi$ may result in a different sequence of actions and states, and hence in a different $\tau$. The set of all possible executions is denoted by $\mathcal{T}$. A *decision* of a policy $\pi$ in a state $s$ is a pair $\langle s, \pi(s) \rangle$. Note that $\pi(s)$ is the learned probability distribution from which an action in this state is obtained; in a deterministic policy, $\pi(s)$ is a single action for each $s$.

**Passing and failing executions** An execution is either successful or failed. We define the success of an execution as a (binary) value of a given assertion on this execution. For example, the assertion can be that the agent reaches its destination eventually, or that the reward of this execution is not lower than $0.75$ of the maximal reward for $\pi$. The assertion induces a Boolean function $C : \mathcal{T} \to \{0, 1\}$. We say that an execution $\tau$ is a *pass* if $C(\tau) = 1$, and is a *fail* otherwise. We use a binary condition for simplicity, as SBFL is designed to work with passing and failing executions. This condition can be relaxed [5], and we plan to investigate generalising this in future work.

**Mutant executions** We use SBFL to understand the impact of replacing actions by a *default action*. The choice of the default action $d$ is context dependent and can be configured by the user. For example, an obvious default action for navigation is "proceed in the same direction". The default action can in principle be as basic as a single action, or as complex as a fully-fledged policy. In our experiments, we evaluate two default actions. The first is "repeat the previous action", defined as:

$$d(s_0, \ldots, s_i, a_0, \ldots, a_{i-1}) = a_{i-1}, \tag{3}$$

and the second is "take a random action". For $A$ being the action space,

$$d(s_0, \ldots, s_i, a_0, \ldots, a_{i-1}) = a_{s_i} \sim Unif(A). \tag{4}$$

Once $a_{s_i}$ has been sampled, it is not re-sampled if the state $s_i$ is revisited; the same action is used. Using "take a random action" as the default action can be useful in cases where there is no other obvious default action. However, we generally expect this to be a worse option than a default action tailored to the environment. As the choice of $d$ depends on the context and the user's goals, it is ultimately a user choice.

Using the default action, we create *mutant executions*, in which the agent takes the default action $d$ whenever it is in one of the *mutant states*. More formally, given a set of mutant states $S_M$, we act according to the policy $\pi_{S_M}$ defined as:

$$\pi_{S_M}(s_0, \ldots, s_i, a_0, \ldots, a_{i-1}) = \begin{cases} \pi(s_i) & s_i \notin S_M, \\ d(s_0, \ldots, s_i, a_0, \ldots, a_{i-1}) & s_i \in S_M. \end{cases} \tag{5}$$

Decisions made in states in which the default action is a very good option are deemed less important. In these states, not following the policy would be less consequential.

## 3.2 Generating the test suite and mutant executions on-the-fly

The naïve approach to generating a comprehensive suite of mutant executions for applying SBFL would be to consider all possible sets of mutant states $S_M$—that is, we would need to consider all possible subsets of the state space $S$. However, the state space of most RL environments is too large to enumerate, and enumerating all possible subsets of $S$ is intractable even for simplistic environments.

We use two algorithmic techniques to address this problem: (a) we generate mutant executions *on-the-fly*, and (b) we use an abstraction function $\alpha : S \to \hat{S}$ to map the full set of states $S$ to a smaller, less complex set of abstract states $\hat{S}$. Examples of these are in the supplementary material, and may for example include down-scaling or grey-scaling images that are input to the agent, or quantising a continuous state space. The set of possible mutations is then the set of subsets of $\hat{S}$, instead of the set of subsets of $S$. We then score the abstract states, rather than the full state space. The test suite of mutant executions produced this way for $\pi$ is denoted $\mathcal{T}(\pi) \subseteq \mathcal{T}$.

**On-the-fly mutation** We maintain a set $S_M \subseteq \hat{S}$ per execution. We begin each execution $\tau$ by initialising the set $S_M$ with the empty set of states. At each step of the execution, upon visiting a state $s$, we check the current $S_M$. If $\alpha(s) \notin S_M$, we add $\alpha(s)$ to $S_M$ according to the predefined *mutation rate* $\mu$ (and take the default action); otherwise, we use $\pi$ to determine the action in this state. In case we re-visit an (abstract) state, we maintain the previous decision of whether to mutate state $s$ or not. This way, the states that are never visited in any of the executions are never mutated; hence, we never consider "useless mutations" that mutate a state that is never visited. We finish by marking $\tau$ as pass or fail according to $C(\tau)$. Note that a mutant execution may visit states not typically encountered by $\pi$, meaning that we are even able to rank states that are out of distribution. This is especially important when trying to understand how the policy behaves in parts of the environment it is unfamiliar with.

Overall, our algorithm has five (tunable) parameters: the size of the test suite $|\mathcal{T}(\pi)|$, the passing condition $C$, the default action $d$, the mutation rate $\mu$ and the abstraction function $\alpha$.

SBFL benefits from a balanced test suite of passing and failing executions [31], a ratio largely determined by the choices of $\mu$ and $C$. The choice of $C$ depends on the context. In our experiments in Section 4.3, our goal was to find the states with highest impact on the expected reward. We set the condition to be "receive more than $X$ reward" for some $X \in \mathbb{R}$, and chose $X$ to yield a balanced suite (i.e. if there were too many failed runs, we lowered $X$). In most cases actions important for achieving at least $X$ reward are important for maximising reward in general, so we found that this worked well. In the cases where some actions are not important for achieving $X$, but later important for getting even higher rewards (e.g., states that only appear after achieving $X$ reward), we would not have considered them important. The choice of $\mu$ is also significant. If it is too large, executions fail too often, and the behaviour in mutant executions is uninteresting. If $\mu$ is too small, we do not mutate states enough to learn anything, and in larger environments fail to mutate many of the states we encounter. In our experiments, we selected $\mu$ manually, but this could easily be automated by a parameter tuning algorithm.

## 3.3 Computing the ranking of the policy decisions

We now explain how to rank states according to the importance of policy decisions made in these states, with respect to satisfying the condition. The ranking method is based on SBFL as described in Section 2.2. We first create the test suite of mutant executions $\mathcal{T}(\pi)$ as described above. We denote the set of all abstract states encountered when generating the test suite $S_{\mathcal{T}} \subseteq \hat{S}$; these are the states to which we assign scores. Any unvisited state is given the lowest possible score by default.

Similarly to SBFL for bug localisation, for each state $s \in S_{\mathcal{T}}$ we calculate a vector $\langle a_{ep}^s, a_{ef}^s, a_{np}^s, a_{nf}^s \rangle$. We use this vector to track the number of times that $s$ was unmutated ($e$) or mutated ($n$) on passing ($p$) and on failing ($f$) executions, and we update these scores based on those executions in which the state was visited. In other words, the vector keeps track of success and failure of mutant executions based on whether an execution took the default action in $s$ or not. For example, $a_{ep}^s$ is the number of passing executions that took the action $\pi(s)$ in the state $s$, and $a_{nf}^s$ is the number of failing executions that took the default action in the state $s$.

**Definition 1** (Ranking). *Given an SBFL measure $m$ and a vector $\langle a_{ep}^s, a_{ef}^s, a_{np}^s, a_{nf}^s \rangle$ for each (abstract) state $s$ in the set $\hat{S}$ of (abstract) states of the policy, we define the ranking function $rank : \hat{S} \to \{1, \ldots, |\hat{S}|\}$ as the ordering of the states in $\hat{S}$ in the descending order according to the values $m(s)$; that is, the state with the maximal value will be the first in the ranking.*

## 4    Experimental evaluation

### 4.1    Research Questions

Our goal is to demonstrate the applicability of our ranking method to a variety of standard environments and to provide evidence of the utility of the generated ranking. We aim to answer the following research questions:

**RQ1:**  How can we measure the relative importance of decisions for achieving the reward? What is a good proxy for measuring this?

**RQ2:**  Does the approach we present in this paper scale to large policies and complex environments?

We answer these questions in Section 4.3 by performing extensive experiments with various environments and policies. In Section 4.4, we discuss possible applications of our ranking (including interpretability), and the effect of choosing a different default action.

### 4.2    Experimental setup

We experimented in several environments. The first is *Minigrid* [7], a gridworld in which the agent operates with a cone of vision and can navigate many different grids, making it more complex than a standard gridworld. In each step the agent can turn or move forward. We also used *CartPole* [4], the classic control problem with a continuous state-space. Finally, to test our ability to scale, we ran experiments with *Atari games* [4].

We use policies that are trained using third-party code. No state abstraction is applied to the gridworld environments (i.e., $\alpha$ is the identity). The state abstraction function for the CartPole environment consists of rounding the components of the state vector between 0 and 2 decimal places, and then taking the absolute value. For the Atari games, as is typically done, we crop the game's border, grey-scale, down-sample to $18 \times 14$, and lower the precision of pixel intensities to make the enormous state space manageable. Note that these abstractions are not a contribution of ours, and were primarily chosen for their simplicity. For our main experiments, we use "repeat previous action" as the default action.

Examples of some environments, and important states found in them, are given in Figs. 1a and 1b. Details about the state abstraction functions, policy training, hyperparameters, etc., are provided in the full version of this paper [22].

We define two further measures in addition to the SBFL ones in Eq. (2), for comparison. Eq. (6a) measures how frequently the state was encountered in the test suite. Eq. (6b) is a random ranking of the states visited by the test suite. We use the FreqVis measure as a baseline because we are not aware of any previous work with the same goals as ours for ranking policy decisions, and a naïve approach to determining importance may be simply looking at how frequently a state is visited.

$$\text{FreqVis:  } a_{ep}^s + a_{ef}^s + a_{np}^s + a_{nf}^s \qquad \text{(6a)} \qquad\qquad \text{Rand:  } \sim \text{Unif}(0, 1) \qquad \text{(6b)}$$

### 4.3    Experimental results

**Performance of pruned policies**    The precise ranking of decisions according to their importance for the reward is intractable for all but very simple policies. To answer **RQ1**, in our experiments, we use the performance of *pruned policies* as a *proxy* for the quality of the ranking computed by our algorithm. In pruned policies, the default action is used in all but the top ranked states. For a given $r$ (a fraction or a percentage), we denote by $rank[r]$ the subset of $r$ top-ranked states. We denote by $\pi^r$ the pruned policy obtained by *pruning* all but the top-$r$ ranked states. That is, an execution of $\pi^r$ retains actions in the $r$ fraction of the most important states from the original policy $\pi$ and replaces

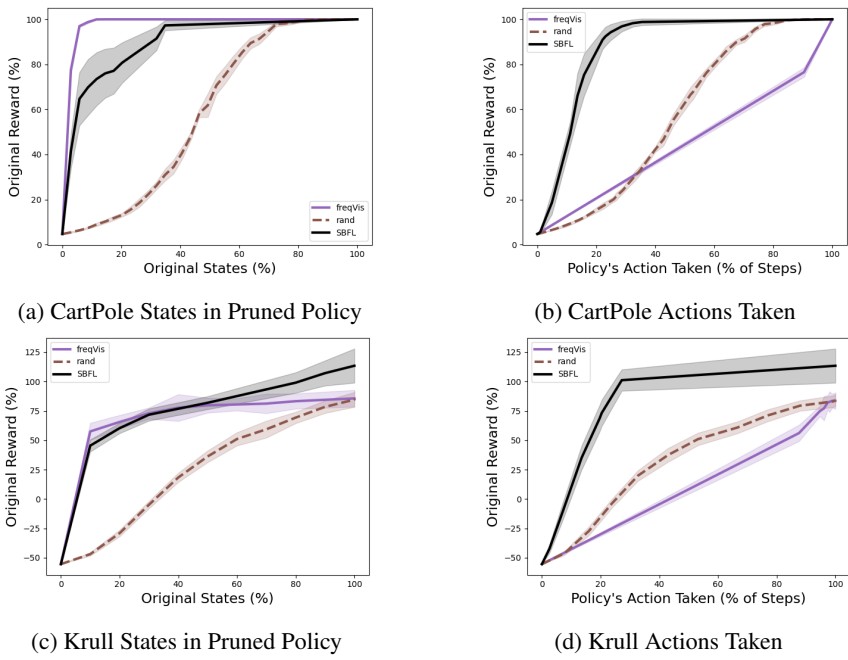

(a) CartPole States in Pruned Policy

(b) CartPole Actions Taken

(c) Krull States in Pruned Policy

(d) Krull Actions Taken

Figure 2: Performance of the pruned policies, measured as a percentage of the original reward. **(a)**,**(c)**: The $x$-axis is the % of states where the original action is taken out of the set of all states encountered in the test suite. **(b)**,**(d)**: The $x$-axis is the expected % of steps in which the original policy is followed over the default action during an execution of the pruned policy. See the supplementary material for more detail.

the rest by default actions. The states that are in $rank[r]$ are called the *original states*. We measure the performance of the pruned policies for increasing values of $r$ relative to the performance of the original policy $\pi$.

These results are given in first four columns of Tab. 1, and some are represented graphically in Figs. 2a,c. In these, SBFL stands for the *SBFL portfolio*, i.e., the combination of the four measures in Eq. (2), where the best result is taken at each point. The results show that the pruned policies obtained using the SBFL ranking can achieve performance that is comparable to $\pi$ with less than $40\%$ of the original decisions (and in some cases the number is as low as $20\%$). The performance of SBFL-based pruning is compared with random pruning in Eq. (6b) and FreqVis pruning in Eq. (6a). At first glance, it seems that the FreqVis pruned policies perform well in Figs. 2a,c. To better understand this, we show in the last two columns of Tab. 1 and in Figs. 2b,d how performance evolves with the proportion of steps in which the original policy is used over the default policy (i.e. not replaced by the default action). As shown in Figs. 2b and 2d, FreqVis does much worse by this metric, because it yields a pruned policy that prefers to use the original policy as often as possible.

We observe that using our ranking method enables a significant pruning of the policies, while maintaining performance. To answer **RQ2**, we experimented with larger and more complex Atari environments. The results demonstrate that our framework is reasonably scalable, but also that the quality of the ranking is significantly improved by using a state abstraction. The results in the larger and more complex Atari environments show the effect of using a test suite that is too small: many states are not encountered during the computation of the ranking. This means that $rank[1]$, in which the original policy is used in *all* of the states discovered while making the ranking, does not recover the performance of the original policy. Increasing the size of the test suite would help, but this is not a scalable solution. Instead, we use better *state abstractions*, which reduce the state space. In CartPole, this allows us to tackle a continuous domain. In the Atari games, even the generic abstraction we use in all the games is sometimes not enough ("x" in Tab. 1).

To show the potential utility of specialised abstractions, we create one for *Breakout* in which we extract the coordinates of the ball and paddle. The results obtained with this abstraction are given in Tab. 1 in the row labelled "Breakout (abs)". While the new abstraction does worse in terms of states

Table 1: Minimum percentage of original states in pruned policies, and percentage of steps in which the original policy is used, before recovering 90% of original performance. Using default action "repeat previous action". Results are reported for the SBFL portfolio ranking and the random ranking. "x" denotes that the required reward was never reached, cf. Sec. 4.3.

| Environment | % of original states restored | | % of steps that use $\pi$ | |
| --- | --- | --- | --- | --- |
| | SBFL | random | SBFL | random |
| MiniGrid | **49 ± 00** | 99 ± 00 | **76 ± 00** | 98 ± 01 |
| Cartpole | **31 ± 04** | 65 ± 02 | **22 ± 04** | 69 ± 02 |
| Alien | x | x | x | x |
| Assault | **45 ± 07** | 100 ± 00 | **93 ± 01** | 100 ± 00 |
| Atlantis | **50 ± 00** | 100 ± 00 | **99 ± 00** | 100 ± 00 |
| BankHeist | x | x | x | x |
| BattleZone | **30 ± 00** | 86 ± 07 | **84 ± 07** | 84 ± 08 |
| Berzerk | **47 ± 12** | 100 ± 00 | **88 ± 03** | 100 ± 00 |
| Boxing | x | x | x | x |
| Breakout | **10 ± 00** | 100 ± 00 | **54 ± 00** | 99 ± 01 |
| Breakout (abs) | **40 ± 00** | 85 ± 05 | **41 ± 00** | 81 ± 06 |
| ChpperCmmnd | x | x | x | x |
| DemonAttack | **20 ± 00** | 99 ± 01 | **98 ± 01** | 99 ± 02 |
| Hero | **48 ± 04** | 96 ± 03 | **86 ± 08** | 96 ± 04 |
| IceHockey | **65 ± 20** | x | **91 ± 08** | x |
| Jamesbond | **30 ± 13** | 68 ± 06 | **59 ± 20** | 67 ± 06 |
| Krull | **75 ± 12** | 99 ± 01 | **35 ± 21** | 98 ± 02 |
| Phoenix | **30 ± 00** | 92 ± 07 | 97 ± 00 | **92 ± 07** |
| Pong | **21 ± 03** | 79 ± 03 | **42 ± 01** | 78 ± 03 |
| Qbert | **40 ± 00** | 100 ± 00 | **84 ± 04** | 100 ± 00 |
| Riverraid | **95 ± 05** | 100 ± 00 | **99 ± 01** | 100 ± 00 |
| RoadRunner | x | x | x | x |
| Seaquest | **48 ± 04** | 94 ± 04 | 92 ± 04 | **91 ± 05** |
| SpaceInvaders | **30 ± 00** | 100 ± 00 | **93 ± 00** | 100 ± 00 |
| StarGunner | **40 ± 00** | 100 ± 00 | **99 ± 00** | 100 ± 00 |
| YarsRevenge | x | x | x | x |

pruned within the policy, it allows us to reach 90% of the policy's original performance with 13% fewer steps in which we use $\pi$ over the default action.

## 4.4 Discussion

**SBFL ranking for better understanding the RL policy**   Any strong claim about the ranking's application to interpretability requires a user study, which is out of scope of this paper. However, we can look to existing research looking at the usefulness of SBFL. While some studies suggest that the users typically do not go over the list of possible causes generated by SBFL linearly (and hence question the usefulness of the ranking) [21], a recent large-scale study demonstrates statistically significant and substantial improvements for the users who use an SBFL tool, and the results hold even for "mediocre" SBFL tools [32]. Based on this evidence, we suggest that the ranking can be used to explain policy decisions, as the ranked list itself would be helpful to identify the most important decisions. In addition, the pruned policies that we construct are simpler than the original policies while achieving a comparable performance, which can make identifying problems more straightforward. Examples are presented in the CartPole and Minigrid sections of the website. Finally, in Fig. 3, we show a heatmap of the scores of each state of a minigrid environment. In each grid square, the agent can be facing four directions. We show the score based on the tarantula measure for these directions from blue (lowest) to red (highest). We show this only for the states visited along a path to the goal. Our heatmap gives information about the general behaviour of the policy. In this case, points at which the agent needs to turn are more red (more important) than points where the agent is walking in a straight line. To understand why the downward-facing states in the right-most column are considered important, refer to our website for a full heatmap and explanation.



Figure 3: Example of a heatmap made based on scores. Colour from least to most important are blue, white and red. **(a)** Our heatmap based on the Tarantula measure, showing all the states an agent encounters while walking to the goal, including the direction in which the agent was facing. For example, in the top left grid location, we show the importance of the state in which the agent is facing right, and the state in which it is facing down.

**A Different Default Action**    Not all environments have an obvious default action. In this case, a straightforward choice is to set the default action to "take a random action". We measured the effect of this choice by running the same experiments with the changed default action, and detailed results are available in the full version of this paper [22]. The results are similar or slightly worse to the ones obtained with the default action "repeat the previous action". In most games, the difference was small ($\leq 10\%$); a few games show a marginal improvement.These results indicate that the choice of a default action has no effect on our conclusions.

**Good vs. bad policies**    Finally, we performed some initial work towards using the ranking to understand the difference between high and low-performing policies. To this end, we produced a ranking according to the high-performing policy, and then compared how the two policies behaved in the highest ranked states. Interestingly, our experiments demonstrate that the high performing and low performing policies agree on $80\%$ of the actions in the top $10\%$ of states. This suggests that the policy training (in CartPole) first picks the 'low-hanging fruit' by performing well in the most important states. The difference between a high-performing and a low-performing policy is mostly in the lower-ranked states. Full results for this experiment are available in the full version of this paper [22].

## 5    Related work

There is a significant body of work on identifying the important parts of trained algorithms, but to the best of our knowledge none suggested to rank the states as we do. Prioritised experience replay [24] looks for the most important *transitions* for training. Saliency maps [10, 29, 33] identify the parts of the state that most influence influence the agent's decision. Sun et al. [26] have previously applied SBFL to visual feature importance for input images given to image classifiers. Other attempts include identifying the important parts of the *representation* of the policy by looking at the parameters of a trained model and pruning it to reduce its size [17]. None of these methods attempt to understand what the important decisions of the policy as a whole are.

Much of the recent work focuses on making deep learning models more interpretable [23, 19, 26]. Many approaches [10, 29, 33] to explaining deep reinforcement learning methods explain the decision made in a single state, without the context of the past or the future behaviour. Iyer et al. [14] explain a single decision via an object-level saliency map by leveraging the pixel-level explanation and object detection. As these methods focus on single decisions, the explanation is typically not sufficient to understand the overall decision-making of the trained agent.

Other work has also attempted to explain entire policies, rather than individual decisions. Ehsan et al. [8] produce natural language explanations for state-action pairs, based on a human-provided corpus of explanations. Topin and Veloso [27] create a Markov chain which acts as an abstraction of the policy, making it easier to reason about the policy. They create the Markov chain by grouping states into abstract states based on how similarly the policy acts in those states; our method is substantially different in that it ranks states based on importance, rather than grouping similar states. While our method allows for the use of abstract states, it is not always necessary, and we are not contributing any specific abstraction function. Similarly, Sreedharan et al. [25] propose a method for creating

a temporal abstraction of a policy by using bottleneck 'landmarks' in the policy's executions. A robot's behaviour can be explained using operator-specified "important program state variables" and "important functions" [12]. We find that the policy-wide decision ranking in this paper is an easier and more general method for understanding the policy. For more work in this vein, we encourage the reader to consult the overview from Chakraborti et al. [6] of the rapidly growing field of Explainable AI Planning (XAIP).

There have been attempts to make more interpretable models, either from scratch [13], or by approximating a trained neural network [28]. In the latter case, our method may be useful for determining in which states the approximation must be most accurate.

## 6 Conclusions

We have applied SBFL-based ranking of states to reinforcement learning and demonstrated that this ranking correlates with the relative importance of states for the policy's performance. The ranking can be used to explain RL policies, similarly to the way ranked program locations are used to understand the causes of a bug. We evaluate the quality of our ranking by constructing simpler pruned policies, where only the most important decisions are made according to the policy, and the rest are default. Our experiments show that the performance of the pruned policies is comparable to the performance of the original policies, thus supporting our claim that the SBFL-based ranking is accurate. Moreover, the pruned policies may be preferable in many use-cases, as they are simpler. Our approach can be scaled with the use of abstractions, as demonstrated by the larger Atari environments.

In the future, we hope to explore different applications of the ranking, new measures, more nuanced hyperparameter selection, and relaxing the binary constraint over the assertion $C$. We do not expect our work to have any negative societal impacts, as it only serves to improve our understanding of the policies we train; the main concern is incorrectly increasing confidence in a policy.

**Funding transparency statement** The authors acknowledge funding from the UKRI Trust-worthy Autonomous Systems Hub (EP/V00784X/1) and the UKRI Strategic Priorities Fund to the UKRI Research Node on Trustworthy Autonomous Systems Governance and Regulation (EP/V026607/1).

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
