# OpenReview forum: "Ranking Policy Decisions"
_NeurIPS.cc/2021/Conference — NeurIPS 2021 Poster_

### Official Review · Reviewer_8qfB · 2021-07-11

**Rating:** 6
**Confidence:** 3

**Summary:**

The goal of this work is to leverage fault localization techniques from software development and debugging as a way to simply learned RL policies. The main premise is that learned policies are unnecessarily complex. In order to simplify their complexity, it is critical to rank the states where the learned policy decisions are critical. Motivated by spectrum-based fault localization, authors leverage it in RL by using similar cost functions to compare complex policy functions to so called mutant executions which are policies that recommend default or previous actions as a way to simplify the complexity of the policy class. Based on such mutant policies (identified on the fly by parametrizing specific aspects), the authors demonstrate they are able to rank important states (based on the policy's impact on outcome). The proposed method is evaluated on standard RL environments (atari, minigrid, cartpole) to quantify efficacy and visually demonstrate interpretability via saliency maps.

**Ethical Concerns:**

I don't see any immediate ethical concerns of this method.

**Limitations And Societal Impact:**

I don't believe there are negative social implications of the proposed work. The authors do address limitations of their methods sufficiently.

**Main Review:**

1. Vague definition of simple policy can be formalized. If the authors are restricting to stochastic policies, why not formalize the notion of simple vs complex policy functions with something like entropy? I think it will also improve the evaluations done in the experiments section.

2. Worthwhile to consider other mutant policies (instead of just the one of replacing with default previous action) to test generalizability of the contributions. I suggest potentially breaking down the atari experiments to try different mutant policies.

3. What is $\hat{S}$ in reference to an on-the-fly mutation (last para, pg 4, line 148). Are you suggesting that abstract state representations are used in combination with on-the-fly mutations to collect trajectories?

4. In experiments, it is unclear why the same set of environments do not achieve the desired reward with either the proposed or the random policy. Can the authors comment on whether it is an issue for the environment or something else?

5. The saliency map comparison is odd because semantically they're comparing very different things depending on the goal. It adds to the confusion to analyze interpretability of the policies. Overall another measure like entropy might be a better motive to formalize what this work is trying to do.

6. From the four different measures used for ranking in states in SBFL, I really would've liked to see more insight into what to use versus not. Instead the authors choose the best performing for every environment(?) which is not adding much insight. Since you've already experimented with all measures, can all measures be reported and discussed?

Overall I am happy to reconsider the score if these concerns are addressed.

----------------------------------------------------------------------------------------
I have gone through author responses and I have increased my score to 6. I trust that authors will make all the changes that came up during the discussions.

**Time Spent Reviewing:**

1.5 hrs

---

> ### Author Response · Authors · 2021-08-10
> **Initial Reply to Reviewer 8qfB**
>
> We greatly appreciate the reviewer’s comments. Below we address their remaining concerns. In general, they advocate for several additional analyses (adding measure of entropy, looking at more default actions, looking at the differences between ranking functions, etc.) We are happy to see the reviewer’s enthusiasm for the questions our work raises. It is easy to add these experiments in our framework, following the reviewer’s suggestions, and we will add them in the final version.
>
> 1. While we emphasize simplicity in this paper, it is not actually a necessary part of the process. In theory, the ‘default action’ can be as complex as an entirely different policy. We focused on conceptually simple default actions here because simple actions are a natural choice for a proof-of-concept work. Our proxy definition of complexity of a pruned policy is, then, the number of decisions it takes according to the original policy. The simplest pruned policy would be the one that always takes the default action, and the most complex one is the original policy. Figures 2b&2d illustrate this measure of complexity, by showing performance as a function of what proportion of the actions taken were from the original policy. This measure captures our intuition regarding the complexity of policies, in contrast to entropy, that is a generic measure. Having said that, it would be interesting to compare the results with the ones obtained from the entropy measure.
> 2. As mentioned in the response to Reviewer ezL2, we will add experiments using random actions as the default action in the camera-ready version. While looking at even more default actions would be interesting, we believe that the results shown in the paper are already demonstrating the potential of our method.
> 3. Yes, that is exactly right. When using a state abstraction, we rank the abstract states, rather than the states themselves. As a result, we mutate at the level of abstract states during the on-the-fly mutation.
> 4. The reason is that we define 100% of states as the set of states seen during ranking (the overall number of states might be infinite). Given the sheer number of states, not all states are seen during the ranking process, and it is likely that new states are encountered during evaluation; hence the discrepancy in the performance. We apologize for the confusion and will clarify this in the camera-ready version.
> 5. Thank you for the suggestion, we will add a comparison to the entropy measure.
> 6. There is a great deal of literature looking at the question of which measure is superior in SBFL (please see the overview in Sec. 2.2). It is well known, and is also observed in our experiments (both in the paper, and the additional ones on the website), that there is no clear overall winner, and different measures perform best in different applications. Hence, the state-of-the-art tools use a portfolio of several measures, which is also what we are using in this paper (we use four measures).

---

> > ### Comment · Reviewer_8qfB · 2021-08-18
> > **Follow-up questions**
> >
> > Thank you to the authors for their clarifications. I feel like I am satisfied with points 1, 2, 3, and 6 I mentioned. Regarding 4 and 5, I have follow-up questions. For 4, you said it is likely that new states that couldn't be ranked have been encountered in evaluation, and I just want to see some evidence of it and confirm that other issues aren't really the reason for this failure. This is of course quite likely in to happen in practice and based on your observations in experiments it is worthwhile adding some discussion around how to deal with this scenario in practice.
> >
> > Re 5, I would like to see a more semantically comparable interpretation and the number of decisions sounds like a more valuable one than saliency map. If you can have thoughts on how this might be presented to a user for interpretability, I would like to see such visualizations. I think it will improve the paper and your main claim if complexity is not the key issue you are trying to tackle.  I would be happy to revisit the scores based on these follow up discussions.

---

> > > ### Author Response · Authors · 2021-08-19
> > > **Addressing Follow-up Questions**
> > >
> > > **Follow up 4.** We are happy to provide evidence for this. Firstly, the only difference in the code used to run the baseline experiment and the pruned policy experiment at 100% is the behaviour on states not seen during the ranking procedure. In the baseline, the original policy is used; in the pruned policy experiment, the default action is used. Secondly, we have looked more closely at the experiment we performed on the atari game Boxing, a particularly extreme example where <30% of the original reward is recovered. We checked what percentage of the encountered states are new states (not seen during the ranking procedure), when using the pruned policy at 100%. As expected, this number is very high, at 75%.
> > >
> > > We agree that this is likely to happen in practice, and there are several possible solutions. Firstly, the ranking procedure can be run for longer and possibly with a higher mutation rate (such that it covers more of the state space). This can be defined as a parameter for the execution, where the program periodically checks how much of the state space has been covered and decides whether to continue or not. Secondly, if the state space is too large to make the first method feasible, then we can apply an abstraction to reduce the state space. Note that this is the approach we adopted in the Cartpole example. By varying the coarseness of the abstraction, we can control the size of the state space. Of course, the tradeoff is that some information from the individual states is lost, and hence it might result in less accurate ranking, especially in very complex environments. Thirdly, we can redefine the pruned policies as using the original policy in any states not visited during the ranking procedure, to ensure that the pruned policy’s performance never drops too dramatically below the original policy’s one. This comes at the cost of using the original policy more often, but may be important in safety-critical scenarios.
> > >
> > > **Follow up 5.** We agree with the reviewer that the number of decisions taken is an intuitively useful tool for interpretability, and we have several visualisations in this spirit. Figure 1b aims to show this by displaying states where the original policy’s decisions were used in red, and default actions in blue. The minigrid and cartpole sections of the [website](https://sites.google.com/view/ranking-policy-decisions/home?authuser=0) show some animated side-by-side comparisons of the original policy and pruned policies, with the same red/blue colour coding. These visualisations show to what extent the pruned policies can simplify behaviour by primarily using the default action. There is also a more detailed version of Figure 1a, showing what our ranking can tell us about the importance of the policy’s decisions for different speeds and pole angles. We will emphasize these in the final paper, as we agree that they are an important part of the discussion.
> > >
> > > **Note for 2.** We have run some preliminary experiments using random behavior as the default action. As the reviewer expressed interest in these results, they may be interested in looking at the table of results in the reply to review ezL2.

---

> > > > ### Comment · Reviewer_8qfB · 2021-08-27
> > > > **Thank you for your response**
> > > >
> > > > My main suggestion at this point would be to just remove the saliency map interpretation, its not quite helpful as a comparison. If you disagree, do let me know why. Also please add these discussions to the draft as they are important for a person who is not familiar with the fault localization literature (i.e. why use a combination of different strategies), causes of consistent failure for certain environments etc. I will increase my score to 6 based on this discussion.

---

> > > > > ### Author Response · Authors · 2021-08-31
> > > > > **Thank you**
> > > > >
> > > > > Thank you for the valuable discussion, which we will be sure to include in the final version.

---

### Official Review · Reviewer_iJVm · 2021-07-13

**Rating:** 8
**Confidence:** 3

**Summary:**

This paper introduces a novel method based on spectrum-based fault localization (SBFL) from the software testing domain to identify the states in which decisions made by an RL policy are most influential in determining whether or not the agent achieves its objective. More specifically, the authors create "mutant trajectories" where, with probability $\mu$, a state is mutated and the action that the agent must take in that state is the "default action" (e.g. "repeat the previous action") or unmutated, where the agent acts according to policy $\pi$ when in that state.  From these mutant trajectories, the authors use SBFL to determine a ranking of states.  SBFL takes as input the vector $\<a_{ep}^{s}, a_{ef}^{s}, a_{np}^{s}, a_{nf}^{s}\>$, where $a_{ep}^{s}$ is the number of successes achieved when state $s$ is a non-mutant state, $a_{ef}^{s}$ is the number of failures achieved when state $s$ is a non-mutant state, and $a_{np}^{s}$, $a_{nf}^{s}$ log the number of successful and failed trajectories when state $s$ is a mutant state.  In order to assess the quality of the ranking returned by SBFL, the authors create "pruned policies" where an agent can only use the policy $\pi$ for the top $r$ states, and must use the default action for all other states.  Overall, in a wide variety of settings ranging from MiniGrid and Cartpole to Atari games, the authors find that, by ranking states using SBFL, they are able to encourage agents to vastly reduce the length of their trajectories while still attaining high reward.  This is a significant contribution from the perspective of creating interpretable AI.

**Limitations And Societal Impact:**

In general, I was very impressed by this paper.  The authors introduced a novel framework to tackle an interesting new problem. The results of applying this framework to a wide variety of tasks were largely impressive.  My only concerns are the following:
- The biggest concern that I have about the widespread adoption of this framework is the fact that there are many parameters that are hard to tune in an automatic way (and, as such, the method fails on some Atari games).  Do you have any suggestions for setting these parameters in an automatic way?  For example, I found it a bit arbitrary how pruned policies were created: namely by keeping top-r.  You could perform a clustering to see if there are 2 clusters of states according to SBFL metrics
- Saturation of Figure 2c and 2d at less than 100% - I may not be understanding something, but I would have expected these to saturate at 100% for freqVis and rand when the original states and policy's actions taken tended to 100%.  Can you explain why this is not the case?
- Concern (Line 155/156): "Note that a mutant execution may visit states not typically encountered by $\pi$, meaning that we are able to even rank states that are typically out of distribution."  The converse is also true though: mutant execution may mean that certain states that are typically encountered by $\pi$ are never reached.  Was this a problem that the authors came across?  If so, did they have any ideas for how to resolve this problem?

**Main Review:**

To the best of my knowledge, the method that this paper introduces is very novel, and is an exciting contribution in itself.  The paper is well written and clear.  The authors conceive of reasonable baselines for comparing their SBFL pruned policies against, despite the fact that the question they tackle here is novel, so there aren't readily available baselines from existing work.  That the authors show the performance of their framework on a wide variety of tasks is a great strength of this paper.

**Time Spent Reviewing:**

3

---

> ### Author Response · Authors · 2021-08-10
> **Initial Reply to Reviewer iJVm**
>
> We are very glad that the reviewer has clearly understood our method and experiments very well, and we are excited to see that they find the paper novel and interesting. We are happy to see that they appreciated the variety of tasks on which we tested our method.
>
> To address the reviewer’s remaining concerns:
> - While it is true that the method has several parameters, whether or not they will require tuning depends on the application. For example, there is often a clear ‘default’ action to use, and if not a random choice is fine (see Response 1) - this is not something that needs tuning. Similarly for the condition ‘C’ against which we test, or the optional abstraction function. The least obvious parameter to tune is the mutation rate \mu. For this, we suggest using a value which balances ‘success’ and ‘failures’ of the mutant execution with respect to C, as SBFL has been observed to work better with balanced datasets (citation [3] in the paper). Standard automatic parameter tuning methods (e.g. binary search) work well. Clustering is a great suggestion, thanks, we will try it as well.
> - The reason is that we define 100% of states as the set of states seen during ranking (the overall number of states might be infinite). Given the sheer number of states, not all states are seen during the ranking process, and it is likely that new states are encountered during evaluation; hence the discrepancy in the performance. We apologize for the confusion and will clarify this in the camera-ready version.
> - While we were concerned this might happen, we did not encounter this problem in practice. One possible solution would be to artificially delay mutations (e.g. set that there will be no mutant states for the first X states visited, and then continue as normal).

---

> > ### Comment · Reviewer_iJVm · 2021-09-02
> > **Thank you for your response**
> >
> > I write to thank the authors for responding to my review.  I was happy with their responses, and continue to recommend acceptance of this paper.

---

### Official Review · Reviewer_3xFH · 2021-07-20

**Rating:** 5
**Confidence:** 2

**Summary:**

This paper studies the problem of "ranking policy decisions," which uses spectrum-based fault localization to compute an offline score for each action,
conditioning on a state. The main idea is that the action with a higher success
frequency should be ranked higher than other actions. Studying the action rank
of policies presents an interesting angle to investigate RL policy from simplifying the DRL policy, interpretability, offline evaluation, while the experiments conducted might not
be sufficient to support either one aspect solidly.

**Limitations And Societal Impact:**

No potential negative societal impact.

**Main Review:**

### Overall

My main concern of this paper is that it tries to cover a range of topics:
pruning DRL policy,  improving DRL interpretation, offline policy evaluation(comparison),
solving any single one of them would make a significant contribution already, while
this work couldn't provide sufficient evidence to demonstrate the proposed method
can work well on any one of those aspects.

### Methodology
What is the reason for accessing a power set of state?
What is the formal definition of "mutant state"?
From (2a) to (2d), we have the formula to rank actions given a state. What is the
formula to rank states as mentioned in Line 107?
As the frequency count relies on the behavior policies used to collect data, it would be better to detail how the trajectories are collected.

### Missing reference
A related work [1] formally learns the rank of actions (decisions in this paper's terminology) in either an online or offline setting.

What is the advantage of the proposed method comparing to learn the rank directly from
data, then applying the learned rank to the downstream task?



### Experiments

Overall, the baseline compared in this paper is relatively weak. Even though the classic
RL algorithm [Q-learning based: DQN, DDQN, etc.][policy gradient based: PPO, A3C, etc.] does not learn the rank of action as their goal, we can use the action value or policy logits learned from those algorithms to serve as a baseline comparison. Based on [1]
the policy gradient approach is learning the listwise rank of actions.

The experiments on the interpretation of RL policy is lack quantitative and
qualitative support. Figure 3 seems not convincing to verify the proposed method is interpretable as any method can plot such a "heatmap".


[1]Lin, Kaixiang, and Jiayu Zhou. "Ranking policy gradient." ICLR (2020).






**Time Spent Reviewing:**

3

---

> ### Author Response · Authors · 2021-08-10
> **Initial Reply to Reviewer 3xFH**
>
> We would like to thank the reviewer for their detailed review.
>
> We stress that our goal is very different from [1]. Their goal is to determine optimal actions to obtain a high-performance policy. By contrast, our goal is to determine in which states the given policy makes its most important decisions. Our algorithm ranks states. The importance ranking of states that we compute in this paper is a key in understanding both the policy and the environment. We will cite [1] in the final version and clarify the difference.
>
> Our responses to the detailed comments are as follows.
> - A mutant execution is an execution in which we replace the policy decisions in some subset of states by the default action. The set of all mutant executions is, therefore, induced by the powerset of the set of all states.
> In the paper, we define a mutant state as a state “in which the agent takes the default action d” (line 129), rather than following the original policy.
> - The equations from 2a to 2d rank states, not actions. These are the equations we use in our ranking algorithm.
> - The point of many of our experiments is to compare the performance of various possible ranking functions; line 107 is not a reference to a specific ranking function. For example, in Figure 2, we compare the SBFL portfolio (described line 235) with FreqVis and random rankings (line 222). On the website we link in a footnote, we show the results for each component of the SBFL portfolio (the equations from 2a to 2d).
> - We are unsure what the reviewer means. We describe in great detail in Section 3 how the trajectories are collected, and they are collected the same way for any of the ranking functions (they are no different for FreqVis than for the SBFL portfolio). The ranking function only comes into play after having collected the mutant trajectories.
>
> As mentioned in other responses, we agree that RQs3-4 are not supported by enough evidence to be answered definitively. Our intention was to open these up for discussion, and point towards interesting first steps in the analysis. However, this is indeed better suited for a discussion section.

---

> > ### Comment · Reviewer_3xFH · 2021-08-25
> > **Response to the authors**
> >
> > Thanks for the response.
> >
> > I misunderstood this work. I went through this work one more time, here is the updated review:
> >
> > The missing reference I mentioned in the original review is not related to this work, and there is no need to cite it.
> >
> > Regarding the problem setting, the motivation of this work is to identify the important states for a better interpretation of the policy.
> > This idea could be an interesting way of interpreting RL.
> >
> > I increased my evaluation for the interesting idea and the extensive experiments across several domains.
> >
> > However, I would not recommend accepting this work as the current version. It would be better to have the paper written in a more formal way following RL literature and addressing the following concerns:
> >
> > - I think the chosen domain is well suited for the repeated action case, Atari has a set of sticky actions which repeat previous actions. When you train the policy, did the authors use sticky_action=True?  If so, the trained policy is able to tolerate repeat action. This approach might not be as general/effective as it seems.
> > - It would be more clear to clarify the important states are ranked based on which policy? Different policies will give the different ranks of states. For example, in Line 273 "in top 10% of states" following which policy. Do different policies agree on the ranking of states? If different policy agrees on the top states, it seems this approach identifies the important states of the environments instead of the policy, which is a bit contradicting to the interpretability of the policy.
> > - Comparing to rank, it seems top-r % of states is more important and the rank among those top states is not used. This means if we alternate the order of states in the top-states, there is no difference in the performance. If this is correct, "identify a minimal set of states" seems more important than "rank of states".
> >
> > The presentation can be improved and avoid some confusion: more appropriate terms can be chosen to fit the RL literature better;
> > e.g., RQ1 as an abbreviation is not commonly used.
> > - Line 227 " top-ranked actions are replaced by the default actions."  means "Following certain policy, the actions in top-ranked states are replaced by the default actions" If I understand it correctly.
> > - A clear mathematical definition of the problem: "ranking policy decision" and the definition of decisions. Sometimes I feel in this paper it refers to "choosing action" while sometimes it seems to refer to "at which state this action is chosen".

---

> > > ### Author Response · Authors · 2021-08-26
> > > **Addressing Reviewer Response**
> > >
> > > Thank you for the insightful comments. We appreciate your positive assessment of the direction and the contribution of the paper. Below we address the comments and requests for clarification in the order they are listed.
> > >
> > > - We are aware of this issue. To prevent this, we used the NoFrameskip-v4 versions of the environments, in which actions aren’t repeated. We will mention this explicitly in the final version.
> > > - This is a good point! In this case study, the states were ranked according to the high-performing policy (we will clarify this in the final version). We then compared the actions selected by each policy in these states. Our goal in this experiment was to investigate why the high-performing policy does better than the low-performing policy. To this end, we looked at the decisions that were most important for the high-performing policy, and checked whether the low-performing policy was aligned with the high-performing one. Surprisingly, we found that in these states, the two policies mostly agreed on which actions to take, suggesting that the difference between the policies is down to more subtle disagreements. The point you are making about identifying important states of the environment is very interesting and plausible, but different experiments are needed to confirm or refute it.
> > > - Our experiments are an abstraction of the approach that adds the original policy decisions according to their ranking one-by-one. Due to the sheer size of the decision space, we instead added the decisions in resolution of 10% in each step. This abstraction ignores the ranking within each 10% bucket, but it certainly exists, and the decisions ranked higher contribute more to the reward than those ranked lower. Furthermore, ranking carries more information than is needed to construct pruned policies. We believe that ranking itself is a valuable contribution, and constructing pruned policies is just an example of one of its applications. We also use the ranking for visualizations, such as the heat map (see Fig.1,3, and the examples on the website).
> > > - We will replace the abbreviation “RQ1” (and similar) by “Research Question 1”.
> > > - Yes this is correct, we will add the clarification you suggested.
> > > - Decisions are pairs <state,P(action)>, where P(action) is the learned distribution over actions from which the policy samples in this state. In a deterministic policy P(action) would be just a single action for each state. Ranking policy decisions ranks the pairs according to their first component (the state). We will add the formal definition and the clarification of the ranking in the final version.

---

### Official Review · Reviewer_ezL2 · 2021-07-26

**Rating:** 6
**Confidence:** 3

**Summary:**

The paper uses an existing ranking mechanism borrowed from software testing to rank the states of trajectories in reinforcement learning. After ranking the states, the authors propose to replace the top performing states with a default decision/action showing that that proposed ranking correlates with the `importance` of the trajectory states. The results show that the performance of the pruned policies (after replacing the states with a pre-defined action) is comparable with the original (non-pruned) policies.

**Limitations And Societal Impact:**

As I think the mutation policy and the abstraction mechanism could have significant impact on the overall ranking and performance of the pruned policies, one of the limitations of this work could be its applicability to more complex environment. Also, some of the claims in the paper are not well studied.

**Main Review:**

The paper seems to tackle an important problem in the reinforcement learning domain, moving towards better understanding of the importance of the decisions made by the RL policies. The paper claims that the proposed ranking mechanism of the RL states followed by mutating the low-performing states with a pre-defined action/decision can still achieve comparable performance to the baseline non-pruned policies. In addition, to tacked the intractability of the states (those that are chosen to be mutant) the authors propose an abstraction mechanism that maps the large state space of RL environments to a simpler and tractable state space. The paper reads well and it seems interesting, however, I feel some of the claims are not sufficiently tested and substantiated.
- If I understand correctly, the order of states also affect the overall performance of policies. That is, if a mutant state occurs at the very beginning of the trajectory that could affect the whole trajectory and may land the agent into a low-performing state. Would you please clarify how the proposed ranking mechanism would deal with this ordering of states? Do you ignore replacing the early states in the trajectory or the ranking mechanism automatically detect these important states?
- Following on the previous question, are there cases where mutating the early states in the trajectory has minimal impact on the overall performance of the policy?
- It seems the mutation policy could have an important impact on the overall performance. That makes me wonder if the proposed approach can be scaled to more complex environment where forming an abstraction mechanism is not straightforward. An interesting result to see is where the mutation is simply `picking a random action`.
-  While I appreciate the authors bringing up RQ3-4, I think the claims are unsubstantiated and not well-supported (as mentioned in the paper, these claims require more thorough study).

**Time Spent Reviewing:**

6

---

> ### Author Response · Authors · 2021-08-10
> **Initial Reply to Reviewer ezL2**
>
> We would like to thank the reviewer for their thoughtful review. In response to their questions:
>
> 1. The reviewer is correct to note that in some cases, an early mutation in a trajectory has a bigger impact on performance. This will depend largely on the environment and the policy being tested. In cartpole, for example, early decisions might not be that important, because the policy can correct for deviations. But if the policy relies on the pole always being close to the center, and is not good at correcting, then an early bad decision will cause more damage (since the pole will fall sooner). Our algorithm does not need to distinguish early and late mutations, as this is covered by the normal ranking procedure - if early mutations lead to bad outcomes, these states will rank highly, because keeping the original policy’s decision in these states is important.
> 2. Such cases do exist, such as the example mentioned in point 1.
> 3. There are two ways of picking a random action. One way is to choose a fresh random action at each step that is to be mutated; we have found that this strategy does not work, as some actions have a much bigger impact than others in the same state, hence the overall ranking does not reflect the real impact of certain actions on the reward (consider, for example, the set of possible actions when crossing a narrow bridge). The second way is to select a default action randomly and fix it as a default for the duration of the experiment. The experiment can then be repeated with a different choice of a default action (again, selected randomly). This is a good strategy for complex environments where a reasonable default action is not obvious. In our experiments, there is always an obvious default action to take.
> 4. We understand the reviewer’s point about RQ3-4, and we agree that the nature of these questions is to initiate a discussion, rather than to have a definitive answer. We will pose these questions as a part of the discussion in the camera-ready version.

---

> > ### Comment · Reviewer_ezL2 · 2021-08-18
> > **Follow-up Question**
> >
> > Thank you for your reply. Some follow-up questions:
> >
> > - Maybe I am missing something here. To clarify, my main concern about the `default action` is that this decision is user-dependent and may not to be extensible to different environment and use cases. Basically, searching for this so-called `default action` could be challenging, especially for the environment with continuous action spaces. To me it seems one of the limitation of this work that would limit its application.
> >
> > - Would you please provide the results for the cases that your approach is not effective or point me to the section that you have provided these results? In addition, I think it would be good if you can add the results/comparisons against the case where a random action is taken instead of the user-defined default action.

---

> > > ### Author Response · Authors · 2021-08-19
> > > **Addressing Follow-up Questions**
> > >
> > > - We agree that in some cases the choice of a default action is not obvious. In this case, we suggest defining "take a random action" as the default action. Following your question, we have conducted experiments with this choice for Minigrid and Cartpole, and the results are shown in the table below. The table is similar to Table 1 in the paper, with the results and a comparison to a random ranking. As expected, the pruned policies are slightly worse, but still perform much better than with a random ranking. This result is to be expected, as a random action is less helpful than “repeat previous action” in these examples, and so it becomes more important to use the original policy’s decisions. As mentioned in other responses, we will add results for the full set of benchmarks in the final version.
> > > | Environment | % of original states restored in policy | % of steps that use π over default action |
> > > | ----- | ----- | ----- |
> > > | Minigrid | 45 (75) | 70 (85) |
> > > | Cartpole | 40 (65) | 60 (70) |
> > > - While we agree that continuous action spaces are more complicated, we believe that there can still be intuitive default actions; for example, “repeat previous action”, or “take constant action *A*” remains a valid choice (if the action space does not change throughout time).
> > > - There are several instances in which our approach was not effective. In Table 1, any result where X is written indicates that the pruned policy never recovered 90% of the original policy’s performance. More detailed results can be found on our website, which is referenced in the paper [website](https://sites.google.com/view/ranking-policy-decisions/atari-games). For example, with Boxing the pruned policy never reaches even 30% of the original policy’s performance. In this case, it is because the state space for Boxing is varied enough that we still see a large number of new states (not seen during ranking) during evaluation. In unranked states, the default action is used, which on its own is usually worse than the original policy. In Boxing, we find that 75% of the states visited at evaluation time had not been seen before. We invite the reviewer to see the discussion of point 4 with reviewer 8qfB, or the second point of reviewer iJVm, for more detail.

---

### Decision · Program_Chairs · 2021-09-27

**Decision:**

Accept (Poster)

**Comment:**

The direction of the paper is based on the premise that learned RL policies are unnecessarily complex and only a small subset of the overall set of decisions yields considerable improvements over some simple baseline. Thus the paper considers ranking of RL policy decisions, and in particular, identifying states where the RL policy's decisions most influence the outcome of the agent achieving its objective. The paper presents a novel method based on the software testing domain, spectrum-based fault localization (SBFL).  The paper uses this to simplify the policy and shows that their approach can help agents considerably reduce the length of their trajectories while still attaining high reward in various settings.


The paper tackles an important problem with a novel approach. Their results show a vast improvement over a variety of settings, however, some concerns are raised by reviewers about the generality especially to complex environments / continuous settings. There are also concerns raised about the interpretability part and RQ3 and 4.


While the paper is far from perfect, in my opinion, the novel approach to an important problem and the empirical results for the settings studied push it over the bar. The authors, however, should carefully revise the paper accommodating all of the points that arose in the reviews, including but not limited to mentioning the use of NoFrameskip-v4 versions of the environments in which actions aren’t repeated, a clarification (ore even better, a formal definition) of the ranking, and addressing the saliency map interpretation.